# Exploring the Collaborative Advantage of Low-level Information on Generalizable AI-Generateted Image Detection

## Abstract

This paper investigates the generalization issue in AI-Generated image detection, aiming to generalize from training on one AI-Generated image dataset to detecting unseen AI-Generated images. Many methods consider extracting low-level information from RGB images to aid the generalization of AI-Generated image detection. However, these methods often consider a single type of low-level information and this may lead to suboptimal generalization. In our analysis, different low-level information often exhibit generalization capabilities for different forgery types. Additionally, simple fusion strategies are insufficient to leverage the detection advantages of each low-level and high-level information for various forgery types. Therefore, we propose the **A**daptive **L**ow-level **E**xperts **I**njection (**ALEI**) framework. Our approach introduces Lora Experts to enable the transformer-based backbone to learn knowledge from different low-level information. We incorporate a Cross-Low-level Attention layer to fuse these features at intermediate layers. To prevent the backbone from losing modeling capabilities for different low-level features, we develop a Low-level Information Adapter that interacts with the features extracted by the backbone. Finally, we propose Dynamic Feature Selection to maximize the generalization detection capability by dynamically selecting the most suitable features for detecting the current image. Extensive experiments demonstrate that our method, finetuned on only four categories of ProGAN data, performs excellently and achieves state-of-the-art results on multiple datasets containing unseen GAN and Diffusion methods.

## 1 Introduction

Advanced AIGC technologies, such as GANs (Goodfellow et al., 2014; Karras et al., 2018; 2019; 2020) and Diffusion models (Dhariwal and Nichol, 2021; Gu et al., 2022; Nichol et al., 2022; Rombach et al., 2022), have seen significant progress, raising concerns about misuse, privacy, and copyright issues. To address these concerns, universal AI-generated image detection methods are essential. A major challenge faced by existing detection methods is how to effectively generalize to unseen AI-Generated Images in real-world scenarios. Existing methods (Ojha et al., 2023; Wang et al., 2020), which primarily use RGB images, often focus on content information, leading to overfitting on AIGC-generated fake images in the training set and a significant drop in generalization accuracy on novel AIGC-generated images.

Recent studies have shown that incorporating low-level information, which refers to fundamental signal properties like noise patterns and subtle artifacts inherent in images (Zamir et al., 2020; Zhang et al., 2017), can significantly enhance the generalization of detection model (Tan et al., 2023b; Jeong et al., 2022a;b; Wang et al., 2023b; Liu et al., 2022; Tan et al., 2023a). For instance, LNP (Liu et al., 2022) and NPR (Tan et al., 2023a) achieve state-of-the-art results by leveraging low-level information. LNP extracts the noise pattern of spatial images based on a well-trained denoising model. NPR investigates the upsampling operations in generative models and designs a module to extract the artifacts associated with this upsampling. These methods focus on analyzing and designing specific types of low-level information for detection. However, the diversity of AIGC technologies and the varied nature of low-level features raise two important questions: How do different types of low-level

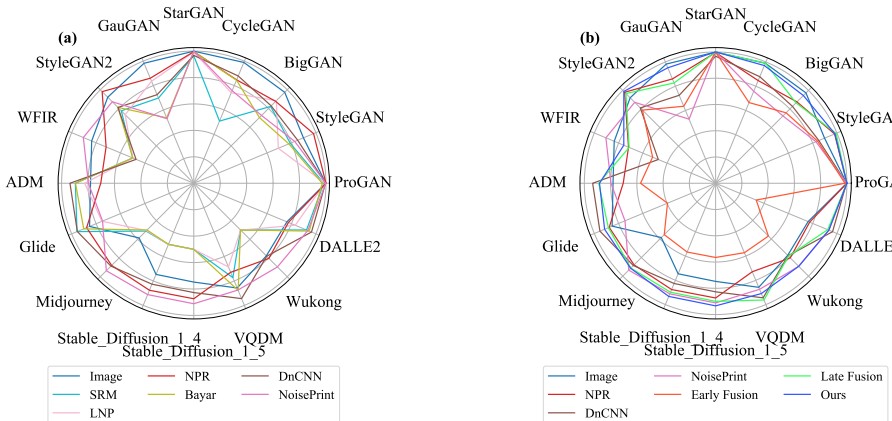

Figure 1: Radar chart of the average accuracy on various forgery test datasets using different low-level features and different fusion strategies.

information contribute to the detection of various AIGC forgeries? Is simply incorporating low-level features into existing models sufficient for optimal detection results?

To address this question, we conducted two sets of analytical experiments. First, we trained detection models using 6 widely used low-level features and evaluated their performance separately on 16 distinct types of AIGC images. We then explored the impact of combining multiple low-level information sources by examining both early and late fusion strategies on these images. The results of these experiments are presented in Fig. 1. Our analysis of these validation experiments yielded two key insights: (a) The effectiveness of different low-level information varies significantly across various types of AIGC image forgeries. (b) Simple fusion mechanisms prove inadequate in fully leveraging these low-level features for optimal detection performance. Thus, it is important to design which and how to integrate low-level features into detection models.

In this paper, we propose the Adaptive Low-level Experts Injection (ALEI) framework, which adaptively incorporates diverse low-level information into the image encoder to effectively detect a wide range of AI-generated image forgeries. Specifically, we train an expert for each type of low-level information using LoRA (Hu et al., 2021) and develop a cross-low-level attention layer to facilitate feature fusion. To address the potential loss of low-level features during deep transformer modeling, we introduce a low-level information adapter. This adapter extracts low-level features through two convolutional layers and maintains ongoing interaction with the backbone's features via our custom-designed injector and extractor. For the final classification, we implement dynamic feature selection and weighting, enabling adaptive utilization of both low-level and high-level features to optimally categorize the current forgery type.

The main contributions of this paper are summarized as follows:

- We provide key insights into the effectiveness of various low-level features for AI-generated image detection, demonstrating that different low-level information generalizes differently across various AIGC forgery types and that simple fusion strategies are insufficient for optimal detection performance.

- We propose the Adaptive Low-level Experts Injection (ALEI) framework, a novel framework that adaptively integrates diverse low-level information into the image encoder. This module includes expert models trained with LoRA, a cross-low-level attention layer, and a low-level information adapter to maintain and effectively fuse low-level features throughout the detection process.

- Experimental results demonstrate that our method achieves competitive performance with state-of-the-art methods across multiple AI-generated image detection benchmark datasets.

## 2 RELATED WORK

### 2.1 AI-GENERATED IMAGE DETECTION

The classification methods of AI-Generated images can be broadly categorized into two main parts: detection based on high-level information and detection based on low-level information.

Following the descriptions in prior works (Triaridis and Mezaris, 2024; Liu et al., 2023b), we refer to the noise patterns extracted from RGB images using carefully designed methods as **low-level information**. In contrast, we define the semantic features extracted from RGB images using deep learning techniques (Liu et al., 2019) as **high-level information**.

**High-level Based Methods.** Early researches utilize images as input and trains binary classification models for GAN-Generated image detection. For instance, (Wang et al., 2020) uses ProGAN-generated images and real images as the training set for a binary classification task, achieving promising results across multiple GAN methods. (Rossler et al., 2019) trains an Xception model to identify deepfake facial images, while (Chai et al., 2020) focuses on detecting recognizable regions within images. More recently, (Ojha et al., 2023) achieves good generalization to diffusion models by fine-tuning the fully connected layers of a CLIP's ViT-L backbone. Building upon this approach, (Liu et al., 2023a) further enhances the detection method's generalization by considering CLIP's text encoding embeddings and introducing frequency-related adapters into the image encoder.

**Low-level Based Methods.** Directly using high-level RGB images as the training set (Wang et al., 2020) often results in limited generalization to AI-generated images outside the training set. Some studies attempt to find universal low-level forgery representations based on high-level images (Luo et al., 2021; Liu et al., 2022; Jeong et al., 2022a; Zhong et al., 2023; Tan et al., 2023b; Wang et al., 2023b; Tan et al., 2023a). (Luo et al., 2021) utilizes SRM filters (Fridrich and Kodovsky, 2012) to extract high-frequency features, enhancing the generalization of face forgery detection. (Jeong et al., 2022a) amplifies artifacts using high-frequency filters to achieve better detection performance. (Liu et al., 2022) extracts noise from images using a denoising network and use this noise for binary classification. (Tan et al., 2023b) classifies gradient maps generated from images using a discriminator pretrained on StyleGAN. (Zhong et al., 2023) trains models based on arrangements of high-frequency features extracted by SRM filters in both adversarial and benign texture regions. (Wang et al., 2023b) utilizes an ADM model for image reconstruction and use the difference between the reconstructed and original images (DIRE) for classification. (Tan et al., 2023a) proposes NPR as a low-level representation of the upsampling process for detection, achieving good generalization across multiple forgery types.

## 2.2 LOW-LEVEL INFORMATION FUSION IN LOW-LEVEL STRUCTURE DETECTION.

Low-level information plays a crucial role in tasks that are difficult for the human eye to perceive. Therefore, many studies explore how incorporating low-level information as input can enhance the performance of methods that use only high-level information. In the field of Camouflaged Object Detection and Image Forgery Detection, (Wang et al., 2023a) guides the detection model to detect camouflaged objects by incorporating depth maps into the detection network based on RGB images. (Guillaro et al., 2023) trains a noise network called Noiseprint using contrastive learning loss to detect image manipulation traces, and then integrate the traces and images into a transformer network for classification and segmentation. (Triaridis and Mezaris, 2024) employs multiple low-level features for adaptive early fusion in the input module of the transformer, achieving state-of-the-art results on multiple datasets. (Liu et al., 2023b) develops a universal framework for detecting various low-level structures. In deepfake detection methods, (Luo et al., 2021; Shuai et al., 2023) introduces high-frequency features using SRM filters (Fridrich and Kodovsky, 2012) through carefully designed fusion modules into the high-level detection branch, applied to facial forgery detection. (Masi et al., 2020) combines RGB and frequency domain information using a two-stream network to detect processed face images and videos. However, in the AI-Generated image detection domain, although many methods emerge using low-level information instead of high-level images for generalization, detection methods that combine multiple low-level information and high-level information remain unexplored.

# 3 ANALYSIS OF LOW-LEVEL INFORMATION

To further investigate the phenomena highlighted in the introduction, we conducted two sets of experiments to analyze the effectiveness of various low-level features and their fusion strategies in AI-generated image detection.

## 3.1 EVALUATION OF INDIVIDUAL LOW-LEVEL FEATURES

*Experimental Setup:* We studied six types of low-level information from different domains: SRM (Fridrich and Kodovsky, 2012), DnCNN (Corvi et al., 2023), NPR (Tan et al., 2023a), LNP (Liu et al., 2022), Bayar (Bayar and Stamm, 2016), and NoisePrint (Guillaro et al., 2023). Following the standard paradigm in the field (Ojha et al., 2023; Wang et al., 2020), we trained on a dataset consisting only of ProGAN and real images, and tested on other AI-generated images using the AIGCDetectBenchmark (Zhong et al., 2023). We utilized the visual encoder of CLIP (Liu et al., 2023a; Ojha et al., 2023) as the backbone, applying LoRA to train the QKV matrix weights in the attention layers. The final classification head was optimized using binary cross-entropy loss.

*Results and Analysis:* The detailed results are presented in Fig. 1 (a) and in the appendix. NPR, DnCNN, and NoisePrint demonstrated better overall performance, showcasing strong generalization capabilities in detecting unseen AIGC images. Image-based methods achieved superior performance on similar GAN datasets but showed limitations on certain Diffusion-based datasets. Different types of low-level information varied in their generalization across different AIGC methods: NPR excelled in detecting mainstream GAN methods, particularly StyleGAN, while DnCNN and NoisePrint performed better on Diffusion-based methods.

*Conclusion:* This experiment supports our hypothesis that although using a single type of low-level information can yield better generalization than RGB images alone, it still results in suboptimal performance across various AIGC forgery types.

## 3.2 EVALUATION OF SIMPLE FUSION STRATEGIES

*Experimental Setup:* To explore the potential of combining multiple low-level information types, we experimented with two simple fusion strategies: (1). Early Fusion: After embedding each input using learnable convolutional layers in the early stages of the backbone, a simple addition operation fuses the inputs. (2). Late Fusion: After extracting features for each input with the backbone, we concatenate the feature vectors and use a learnable classification head for training. Both of the above backbones are trained using LoRA.

*Results and Analysis:* The results are presented in Fig. 1(b) and in the appendix. Early fusion appeared to confuse some key features, leading to a loss of generalization. Late fusion, while showing strong results, still suffered from insufficient utilization, failing to match the generalization of individual low-level information types for certain AI-generated images.

*Conclusion:* Simple fusion mechanisms prove inadequate in fully leveraging these low-level features for optimal detection performance across various AIGC forgery types.

Based on these findings, we propose the Adaptive Low-level Experts Injection (ALEI) framework, specifically designed for AI-generated image detection. Given the superior performance of NPR, DnCNN, and NoisePrint, and following the principle of Occam's Razor, we conducted further experiments using only these three low-level information types. The integration of additional low-level information is also feasible within our framework, which we discuss further in the appendix. The detailed method will be presented in the subsequent sections.

## 4 METHODOLOGY

### 4.1 OVERVIEW

This paper aims to address the issue of generalization in AI-Generated image detection. Given an input image $I \in \mathbb{R}^{H \times W \times 3}$, where $H$ and $W$ denote the height and width respectively, we extract multiple low-level information $C = \{C_1, C_2, ..., C_M\}$, where each $C_i \in \mathbb{R}^{H \times W \times 3}, i = 1, 2, 3, ..., m$. Following UniFD (Ojha et al., 2023), our approach uses the CLIP's visual encoder ViT-L as the backbone in Fig. 2. To enable the model, pretrained on high-level images, to accept various low-level information inputs and ensure effective integration, while avoiding insufficient fusion either in the early or late stages, we transform the original transformer block into a Cross-Low-level Expert LoRA Transformer Block, which will be introduced in Section 4.2. Furthermore, to prevent the loss of low-level input characteristics in deep transformer modeling, we employ a low-level information interaction adapter. This adapter further transmits the features of low-level information into the ViT

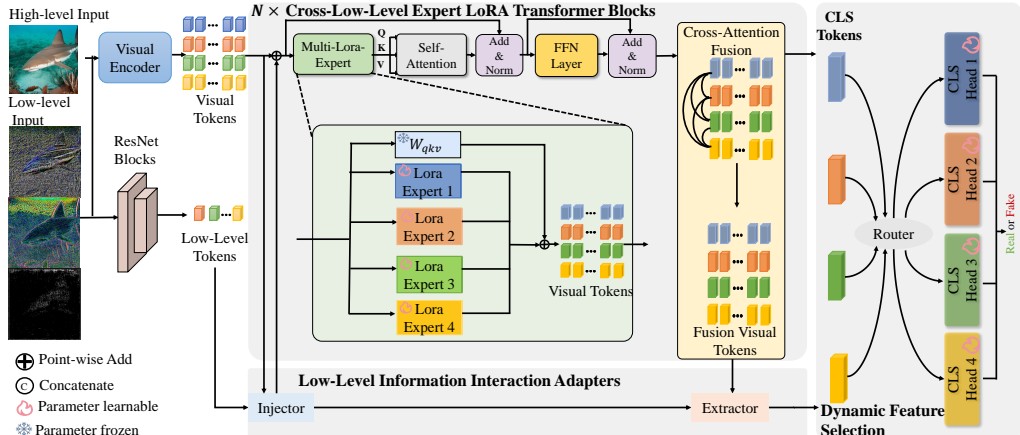

Figure 2: The overall framework of our proposed method. Our method consists of three main components: Cross-Low-level Expert LoRA Transformer Block, Low-Level Information Interaction Adapter, and Dynamic Feature Selection. These modules will be explained in the methods section.

for enhanced interaction, as discussed in Section 4.3. Finally, to select the most suitable features for different types of forgeries, we propose a dynamic feature selection classification head to choose the most appropriate low-level features for the current type of forgery, which will be detailed in Section 4.4. The overall training phase of our framework will be presented in Section 4.5.

## 4.2 CROSS-LOW-LEVEL EXPERT LORA TRANSFORMER BLOCK

In our approach, we avoid merging features from different modalities using straightforward fusion techniques. Instead, we strive to preserve the unique characteristics of each modality while capturing the interactions and influences between them. For the $M + 1$ different low-level inputs with the high-level image input $I$ denoted as $C_0$ and added to the set $C$, $C_j$, $(j = 0, 1, 2, ..., M)$, the visual encoder initially transforms the input tensors of size $\mathbb{R}^{H \times W \times 3}$ into $D$-dimensional image features $F_0^j \in \mathbb{R}^{(1+L) \times D}$, where 1 represents the CLS token of the image, and $L = \frac{H \times W}{P^2}$ with $P$ representing the number of patches. The input features for the $j^{th}$ modality $C_j$ through the $i^{th}$ transformer block are denoted as $F_i^j \in \mathbb{R}^{(1+L) \times D}$, $i = 0, 1, 2, ..., N$, where $N$ denotes the number of blocks in the transformer. The transformer module takes the patch-embedded features $F_0^j$ as input for each low-level information.

Considering the distinctiveness of each modality, we aim to embed the knowledge of each modality into the CLIP visual backbone without affecting the original pretrained weights. We utilize the fine-tuning technique Lora (Hu et al., 2021), widely used in large language models and diffusion models, to introduce modal knowledge through an additional plug-and-play module.

Each block consists of our designed Multi-Lora-Expert Layer in Fig. 3(a), Self-Attention, residual connections, Layer Normalization and an FFN layer. In the Multi-Lora-Expert Layer at layer $i$, we employ Lora to process features specific to each modality by designing different Lora experts. The computation is as follows:

$$\hat{F}_i^{(j)} = W_{qkv} \cdot F_i^{(j)} + \frac{\alpha}{r} \Delta W_j \cdot F_i^{(j)} = W_{qkv} \cdot F_i^{(j)} + \frac{\alpha}{r} B_j A_j \cdot F_i^{(j)} \tag{1}$$

Here, $\hat{F}_i^{(j)}$ represents the output of $F_i^{(j)}$ after processing by the $j^{th}$ Lora expert and we set $r = 4$ and $\alpha = 8$, $W_{qkv}$ denotes the matrix weights of the qkv in the attention layer and $\Delta W_j = B_j A_j$ is the trainable parameter of the $j^{th}$ Lora expert. Next, $\hat{F}_i^{(j)}$ serves as the input for the self-attention $Q, K, V$ in the original CLIP, and the output after the FFN layer is denoted as $\overline{F}_i^{(j)}$. Noting that the features of each modality are computed in parallel without interaction, we employ a cross-modality attention layer in the original output section to facilitate interaction between modalities, as computed by:

$$\overline{F}_i = \text{Concatnate}[\overline{F}_i^{(j)}, \ 0 \le j \le C]$$
$$F_{i+1} = \overline{F}_i + \beta_i \text{MHA}(\text{LN}(\overline{F}_i), \text{LN}(\overline{F}_i), \text{LN}(\overline{F}_i)) \tag{2}$$

Here, LN($\cdot$) represents LayerNorm, and the attention layer MHA($\cdot$) is suggested to use a multi-head attention mechanism with the number of heads set to 4. Furthermore, we apply a learnable vector $\beta_{ik} \in \mathbb{R}^D$ to balance the output of the attention layer with the input features, initially set to 0. This initialization strategy ensures that the unique features of each modality do not undergo drastic changes due to the injection of features from other modalities and adaptively integrates features related to forgery types contained in other modalities.

### 4.3 LOW-LEVEL INFORMATION INTERACTION ADAPTER

Many work (Zhao et al., 2023; Peng et al., 2021; Yuan et al., 2021) suggests that the deeper layers of transformers might lead to the loss of low-level information, focusing instead on the learning of semantic information. Inspired by (Chen et al., 2022), to prevent our framework from losing critical classification features related to forgery types during the fusion of low-level information, we introduce a low-level information interaction adapter. This adapter is designed to capture low-level information priors and to enhance the significance of low-level information within the backbone. It operates parallel to the patch embedding layer of the CLIP image encoder and does not alter the architecture of the CLIP visual encoder. Unlike the vit-adapter, which injects spatial priors, our adapter injects low-level priors.

As illustrated, we utilize the first two blocks of ResNet50 (He et al., 2016), followed by global pooling and several $1 \times 1$ convolutions applied at the end to project the low-level information $C_1, C_2, ..., C_M$ into $D$ dimensions. Through this process, we obtain the feature vector $G_0 \in \mathbb{R}^D$ extracted from the low-level encoder. To better integrate our features into the backbone, we design a cross-attention-based low-level feature injector and a low-level feature extractor.

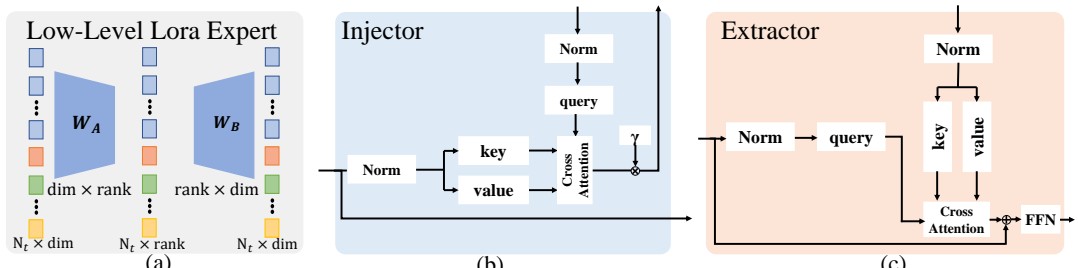

Figure 3: The key components of our method.

**Low-level Feature Injector.** This module is used to inject low-level priors into the ViT. As shown in Fig. 3(b), for the output from each modality feature of the $i^{th}$ block of CLIP using ViT-L, the features are concatenated into a feature vector $F_i \in \mathbb{R}^{(1+M)\cdot(1+L)\times D}$, which serves as the query for computing cross-attention. The low-level feature $G_i$ acts as the key and value in injecting into the modal feature $F_i$, represented by the following equation:

$$\tilde{F}_i = F_i + \gamma_i \text{MHA}(\text{LN}(F_i), \text{LN}(G_i), \text{LN}(G_i)) \tag{3}$$

As before, LN and MHA operations respectively represent LayerNorm and multi-head attention mechanisms, with the number of heads set to 4. Similarly, we use a learnable vector $\gamma_i \in \mathbb{R}^D$ to balance the two different features.

**Modal Feature Extractor.** After injecting the low-level priors into the backbone, we perform the forward propagation process. We concatenate the output of each modality feature of the $(i + 1)^{th}$ block to obtain the feature vector $F_{i+1}$ and then apply a module composed of cross-attention and FFN to extract modal features, as shown in Fig. 3(c). This process is represented by the following equations:

$$\tilde{G}_i = G_i + \eta_i \text{MHA}(\text{LN}(G_i), \text{LN}(F_{i+1}), \text{LN}(F_{i+1})) \tag{4}$$

$$G_{i+1} = \tilde{G}_i + \text{FFN}(\text{LN}(\tilde{G}_i)) \tag{5}$$

Here, the low-level feature $G_i \in \mathbb{R}^D$ serves as the query, and the output $F_{i+1} \in \mathbb{R}^{(1+M)\cdot(1+L)\times D}$ from backbone acts as the key and value. Similar to the low-level feature injector, we use a learnable vector $\eta_i \in \mathbb{R}^D$ to balance the two different features. $G_{i+1}$ is then used as the input for the next low-level feature injector.

## 4.4 DYNAMIC FEATURE SELECTION

As mentioned in the introduction, since different modal features are often sensitive to different types of forgeries, simple feature concatenation or averaging followed by training with a unified classification head might lose some modal advantages for detecting certain types of forgeries. To better integrate modal features for generalizing to various forgery type detections, inspired by the mixed experts routing dynamic feature selection (Shazeer et al., 2017), we introduce a dynamic modal feature selection mechanism at the final output classification feature part of the model. Specifically, we extract the cls tokens of the final output modal features, concatenate them, and denote this as $F_{cls} \in \mathbb{R}^{(1+M)\cdot D}$, which serves as the input for the dynamic router. The dynamic router employs a learnable fully connected neural network, with its matrix parameter defined as $W_{Router} \in \mathbb{R}^{(1+M)\cdot D \times (1+M)}$. The probability distribution for selecting each modal feature is computed as follows:

$$p = \text{SoftMax}(W_{Router} F_{cls}) \tag{6}$$

For each modality, a corresponding classification head $\text{head}_i, i = 0, 1, 2, ..., M$, is prepared. The final classification result $\hat{y}$ is obtained through the following equation:

$$\hat{P}(y) = \sum_{i=0}^{M} p_i \cdot \text{head}_i(F_{cls}^i) \tag{7}$$

Here, $F_{cls}^i$ represents the cls token of the $i^{th}$ modality. By adaptively learning a dynamic modal feature selection module, we enable the selection of the most suitable modal features for integration, thus allowing the classification to be tailored to the forgery type of the current image under detection. To balance the selection of different experts, we use entropy regularization loss as an additional constraint, as shown below:

$$\mathcal{L}_{moe} = -\sum_{i=0}^{M} p_i \log p_i \tag{8}$$

## 4.5 TRAINING PHASE

We first train Lora Expert and the low-level information encoder for each type of low-level information and the high-level image information to ensure that the model learns knowledge relevant to AI-Generated image detection from both low-level and high-level information. Let the true label be $y$ and the model's prediction be $\hat{P}(y)$. The training is performed using the cross-entropy loss as defined in Eq.9. Subsequently, we load these pre-trained weights into our framework and further train our carefully designed fusion module to ensure the adequate and appropriate fusion of each type of low-level and high-level information. Our final fused prediction results are given in Eq.7, and we optimize our overall framework using Eq.10 as well, the loss is composed of the classification loss (Eq.9) and the expert balance regularization loss (Eq.8) weighted together. In our experiments, we set $\lambda = 0.1$.

$$\mathcal{L}_{cls} = -y \cdot \log \hat{P}(y) - (1 - y) \cdot \log(1 - \hat{P}(y)). \tag{9}$$

$$\mathcal{L}_{total} = \mathcal{L}_{cls} + \lambda \mathcal{L}_{moe} \tag{10}$$

# 5 EXPERIMENT

## 5.1 EXPERIMENTAL SETUPS

**Training Dataset.** To ensure a fair comparison, we adhere to the training set proposed by (Wang et al., 2020). Testing is then conducted on other unseen forgery types, such as those generated by different GANs or new diffusion models. This training set comprises 20 different categories, with each category containing 18,000 synthetic images generated by ProGAN. Additionally, an equal number of real images sampled from the LSUN dataset are included. As in previous methods (Jeong et al., 2022a;b; Tan et al., 2023a; Liu et al., 2023a), we restrict the training set to four categories: car, cat, chair, and horse.

**Testing Dataset.** To further evaluate the generalization capability of the proposed method in real-world scenarios, we employ various real-world images and images generated by diverse GANs and Diffusions. The evaluation dataset follows the test datasets proposed by previous methods and primarily includes the following datasets:**CNNDetectionBenchmark (Wang et al., 2020)**, **GANGenDetectionBenchmark (Tan et al., 2024)**, **UniversalFakeDetectBenchmark (Ojha et al., 2023)** and **AIGCDetectBenchmark (Zhong et al., 2023)**. More details about testing dataset are provided in the Appendix.

**SOTA Methods Details.** This paper aims to establish a framework that integrates multiple low-level and high-level features to enhance the generalization capabilities of AI-generated image detection. To this end, we conduct extensive comparisons with several state-of-the-art methods that explore generalization in AI-generated image detection, including: CNNDet (Wang et al., 2020), FreDect (Frank et al., 2020a), Fusing (Ju et al., 2022), GramNet (Liu et al., 2020), Frank (Frank et al., 2020b), Durall (Durall et al., 2020), Patchfor (Chai et al., 2020), F3Net (Qian et al., 2020), SelfBlend (Shiohara and Yamasaki, 2022), GANDet (Mandelli et al., 2022), FrePGAN (Jeong et al., 2022b),BiHPF (Jeong et al., 2022a), LNP (Liu et al., 2022), LGrad (Tan et al., 2023b), DIRE-G (Wang et al., 2023b), DIRE-D (Wang et al., 2023b), UnivFD (Ojha et al., 2023), PatchCraft (Zhong et al., 2023), FAFormer (Liu et al., 2023a), and NPR (Tan et al., 2023a). In this context, DIRE-D refers to the results obtained using the pretrained weights from the original DIRE model, trained on the ADM dataset, while DIRE-G refers to the results obtained from retraining the DIRE model using weights trained on the ProGAN dataset.

**Implementation Details.** Our main training and testing settings largely follow previous research. First, the input images are resized to $256 \times 256$, then center-cropped to $224 \times 224$. During training, we use a random cropping strategy, while for testing, only center cropping is applied. We train our method using the Adam optimizer with parameters (0.9, 0.999), a learning rate of $2 \times 10^{-4}$, and a batch size of 32. Our method is implemented using the PyTorch framework on four Nvidia GeForce RTX 3090 GPUs. The training period is set to 10 epochs. We report the average accuracy (Acc.) and average precision (A.P.) during the evaluation for each forgery type. More details related to our method and baseline methods are provided in the Appendix.

## 5.2 COMPARED WITH SOTA METHODS

**Comparisons on AIGCDetectBenchmark.** Tab. 1 reports results of our method and baseline methods on AIGCDetectBenchmark. Our method outperforms previous state-of-the-art methods by 3.44% across 16 different forgery datasets. This notable achievement is largely due to the generalization capability offered by diverse low-level features for AI-generated image detection, along with the effective integration of low-level information containing various forensic clues. This enables our method to generalize well to unseen fake images using a limited amount of ProGAN training data.

Table 1: The detection accuracy comparison between our approach and baselines. Among all detectors, the best result and the second-best result are denoted in boldface and underlined, respectively. The complete table will be presented in the Appendix.

| Generator | CNNDet | GramNet | LNP | LGrad | DIRE-G | DIRE-D | UnivFD | PatchCraft | Ours |
|---|---|---|---|---|---|---|---|---|---|
| ProGAN | **100.00** | 99.99 | 99.95 | 99.83 | 95.19 | 52.75 | 99.81 | **100.00** | **100.00** |
| StyleGAN | 90.17 | 87.05 | 92.64 | 91.08 | 83.03 | 51.31 | 84.93 | 92.77 | **98.35** |
| BigGAN | 71.17 | 67.33 | 88.43 | 85.62 | 70.12 | 49.70 | 95.08 | **95.80** | 94.51 |
| CycleGAN | 87.62 | 86.07 | 79.07 | 86.94 | 74.19 | 49.58 | **98.33** | 70.17 | 97.03 |
| StarGAN | 94.60 | 95.05 | **100.00** | 99.27 | 95.47 | 46.72 | 95.75 | 99.97 | **100.00** |
| GauGAN | 81.42 | 69.35 | 79.17 | 78.46 | 67.79 | 51.23 | **99.47** | 71.58 | 95.19 |
| StyleGAN2 | 86.91 | 87.28 | 93.82 | 85.32 | 75.31 | 51.72 | 74.96 | 89.55 | **98.88** |
| whichfaceisreal | **91.65** | 86.80 | 50.00 | 55.70 | 58.05 | 53.30 | 86.90 | 85.80 | 75.71 |
| ADM | 60.39 | 58.61 | 83.91 | 67.15 | 75.78 | **98.25** | 66.87 | 82.17 | 88.43 |
| Glide | 58.07 | 54.50 | 83.50 | 66.11 | 71.75 | **92.42** | 62.46 | 83.79 | 91.53 |
| Midjourney | 51.39 | 50.02 | 69.55 | 65.35 | 58.01 | 89.45 | 56.13 | 90.12 | **91.56** |
| SDv1.4 | 50.57 | 51.70 | 89.33 | 63.02 | 49.74 | 91.24 | 63.66 | **95.38** | 93.28 |
| SDv1.5 | 50.53 | 52.16 | 88.81 | 63.67 | 49.83 | 91.63 | 63.49 | **95.30** | 93.38 |
| VQDM | 56.46 | 52.86 | 85.03 | 72.99 | 53.68 | **91.90** | 85.31 | 88.91 | 90.94 |
| wukong | 51.03 | 50.76 | 86.39 | 59.55 | 54.46 | **90.90** | 70.93 | 91.07 | 89.46 |
| DALLE2 | 50.45 | 49.25 | 92.45 | 65.45 | 66.48 | 92.45 | 50.75 | **96.60** | 93.32 |
| Average | 69.73 | 68.43 | 85.28 | 75.11 | 67.90 | 72.70 | 76.80 | 89.85 | **93.29** |

**Comparison on GANGenDetectionBenchmark.** Tab. 2 evaluates the Acc. and A.P. metrics on GANGenDetection, with test results on CNNDetection provided in the Appendix. The test datasets were unseen during training, with ProGAN in the test set comprising 20 classes, compared to only

4 in the training set. Our method outperforms several baseline methods and achieves comparable results to the state-of-the-art methods NPR (Liu et al., 2023a), improving average accuracy by $2.1\%$ and $1.5\%$. This indicates that our method, by incorporating multiple low-level information, enhances detection performance uniformly across various GAN generation methods.

Table 2: Cross-GAN-Sources Evaluation on the GANGenDetection (Tan et al., 2024). Partial results from (Tan et al., 2023a). The complete table will be presented in the Appendix.

| Method | AttGAN | | BEGAN | | CramerGAN | | InfoMaxGAN | | MMDGAN | | RelGAN | | SNGAN | | Mean | |
|---|---|---|---|---|---|---|---|---|---|---|---|---|---|---|---|---|
| | Acc. | A.P. | Acc. | A.P. | Acc. | A.P. | Acc. | A.P. | Acc. | A.P. | Acc. | A.P. | Acc. | A.P. | Acc. | A.P. |
| CNNDet | 51.1 | 83.7 | 50.2 | 44.9 | 81.5 | 97.5 | 71.1 | 94.7 | 72.9 | 94.4 | 53.3 | 82.1 | 62.7 | 90.4 | 62.3 | 82.9 |
| Frank | 65.0 | 74.4 | 39.4 | 39.9 | 31.0 | 36.0 | 41.1 | 41.0 | 38.4 | 40.5 | 69.2 | 96.2 | 48.4 | 47.9 | 47.5 | 54.7 |
| Durall | 39.9 | 38.2 | 48.2 | 30.9 | 60.9 | 67.2 | 50.1 | 51.7 | 59.5 | 65.5 | 80.0 | 88.2 | 54.8 | 58.9 | 60.3 | 63.3 |
| Patchfor | 68.0 | 92.9 | 97.1 | 100.0 | 97.8 | 99.9 | 93.6 | 98.2 | 97.9 | 100.0 | 99.6 | 100.0 | 97.6 | 99.8 | 90.1 | 95.4 |
| F3Net | 85.2 | 94.8 | 87.1 | 97.5 | 89.5 | 99.8 | 67.1 | 83.1 | 73.7 | 99.6 | 98.8 | 100.0 | 51.6 | 93.6 | 75.4 | 93.1 |
| SelfBlend | 63.1 | 66.1 | 56.4 | 59.0 | 75.1 | 82.4 | 79.0 | 82.5 | 68.6 | 74.0 | 73.6 | 77.8 | 61.6 | 65.0 | 65.8 | 69.7 |
| GANDet | 57.4 | 75.1 | 67.9 | 100.0 | 67.8 | 99.7 | 67.6 | 92.4 | 67.7 | 99.3 | 60.9 | 86.2 | 66.7 | 90.6 | 66.1 | 91.6 |
| LGrad | 68.6 | 93.8 | 69.9 | 89.2 | 50.3 | 54.0 | 71.1 | 82.0 | 57.5 | 67.3 | 89.1 | 99.1 | 78.0 | 87.4 | 68.6 | 80.8 |
| UnivFD | 78.5 | 98.3 | 72.0 | 98.9 | 77.6 | 99.8 | 77.6 | 98.9 | 77.6 | 99.7 | 78.2 | 98.7 | 77.6 | 98.7 | 77.6 | 98.8 |
| NPR | 83.0 | 96.2 | 99.0 | 99.8 | 98.7 | 99.0 | 94.5 | 98.3 | 98.6 | 99.0 | 99.6 | 100.0 | 88.8 | 97.4 | 93.2 | 96.6 |
| Ours | **86.2** | **97.8** | **100.0** | **100.0** | **100.0** | **100.0** | **98.6** | **99.9** | **99.3** | **99.8** | **100.0** | **100.0** | **90.4** | **98.7** | **95.3** | **98.1** |

**Comparison on UniversalFakeDetectBenchmark.** Tab. 3 evaluates the Acc. and A.P. metrics on the Diffusions dataset from UniversalFakeDetect. Given that our method is trained on ProGAN, this setting poses a challenge as the fake images originate from different Diffusion methods, which differ significantly from GAN generation processes. Nevertheless, our method exhibits strong generalization capabilities across various Diffusion models. Compared to state-of-the-art methods NPR (Liu et al., 2023a) and FAFormer (Tan et al., 2023a), our method enhances Acc. by $2.0\%$ and $3.4\%$, respectively, and A.P. by $1.7\%$ and $3.6\%$, respectively. These results strongly suggest that the low-level information utilized contains critical clues that generalize well to diffusion detection, resulting in improved performance.

Table 3: Cross-Diffusion-Sources Evaluation on the diffusion test set of UniversalFakeDetect (Ojha et al., 2023). Partial results from (Liu et al., 2023a; Tan et al., 2023a). The complete table will be presented in the Appendix.

| Method | DALLE | | Glide_100_10 | | Glide_50_27 | | ADM | | LDM_100 | | LDM_200 | | Mean | |
|---|---|---|---|---|---|---|---|---|---|---|---|---|---|---|
| | Acc. | A.P. | Acc. | A.P. | Acc. | A.P. | Acc. | A.P. | Acc. | A.P. | Acc. | A.P. | Acc. | A.P. |
| CNNDet | 51.8 | 61.3 | 53.3 | 72.9 | 54.2 | 76.0 | 54.9 | 66.6 | 51.9 | 63.7 | 52.0 | 64.5 | 52.8 | 67.4 |
| Frank | 57.0 | 62.5 | 53.6 | 44.3 | 52.0 | 42.3 | 53.4 | 52.5 | 56.6 | 51.3 | 56.4 | 50.9 | 54.5 | 49.6 |
| Durall | 55.9 | 58.0 | 54.9 | 52.3 | 51.7 | 49.9 | 40.6 | 42.3 | 62.0 | 62.6 | 61.7 | 61.7 | 54.3 | 54.0 |
| Patchfor | 79.8 | 99.1 | 87.3 | 99.7 | 84.9 | 98.8 | 74.2 | 81.4 | 95.8 | 99.8 | 95.6 | 99.9 | 86.8 | 97.2 |
| F3Net | 71.6 | 79.9 | 88.3 | 95.4 | 88.5 | 95.4 | 69.2 | 70.8 | 74.1 | 84.0 | 73.4 | 83.3 | 79.1 | 86.5 |
| SelfBlend | 52.4 | 51.6 | 58.8 | 63.2 | 64.2 | 68.3 | 58.3 | 63.4 | 53.0 | 54.0 | 52.6 | 51.9 | 56.3 | 58.7 |
| GANDet | 67.2 | 83.0 | 51.2 | 52.6 | 51.7 | 53.5 | 49.6 | 49.0 | 54.7 | 65.8 | 54.9 | 65.9 | 54.3 | 60.1 |
| LGrad | 88.5 | 97.3 | 89.4 | 94.9 | 90.7 | 95.1 | 86.6 | 100.0 | 94.8 | 99.2 | 94.2 | 99.1 | 90.9 | 97.2 |
| UnivFD | 89.5 | 96.8 | 90.1 | 97.0 | 91.1 | 97.4 | 75.7 | 85.1 | 90.5 | 97.0 | 90.2 | 97.1 | 86.9 | 94.5 |
| NPR | 94.5 | 99.5 | 98.2 | 99.8 | 98.2 | 99.8 | 75.8 | 81.0 | 99.3 | 99.9 | **99.1** | **99.9** | 95.2 | 97.4 |
| FAFormer | **98.8** | **99.8** | 94.2 | 99.2 | 94.7 | 99.4 | 76.1 | 92.0 | 98.7 | 99.9 | 98.6 | 99.8 | 93.8 | 95.5 |
| Ours | 97.7 | 99.7 | **97.9** | 99.2 | **98.6** | **99.9** | **90.1** | **96.4** | **99.5** | **99.9** | 98.9 | 99.3 | **97.3** | **99.1** |

## 5.3 ABLATION STUDY

We conducted ablation studies to verify the effectiveness of the key components in our method using the AIGCDetectBenchmark, which includes both GAN- and Diffusion-synthesized images, providing a challenging dataset for generalization detection.

**Combination of different low-level information.** To demonstrate the effectiveness of the low-level information used in our method, we compared its performance with different low-level information in Tab.4. Each type of low-level information individually achieved over $83\%$ Acc. and $89\%$ A.P. on the test set, indicating generalization performance on synthetic images. As we progressively added low-level information, performance improved, with an overall enhancement of $8.0\%$ in Acc. and $6.6\%$ in A.P. We visualized features of different low-level information using t-SNE (Van Der Maaten, 2014) plots for various synthetic image methods (StyleGAN, BigGAN, ADM, Stable Diffusion) and the distribution of low-level features for different forgery types in Fig. 4. As noted in the Analysis section, different low-level information provides key clues for detecting synthetic image methods, establishing distinct boundaries. For example, Image and NPR effectively separate BigGAN and

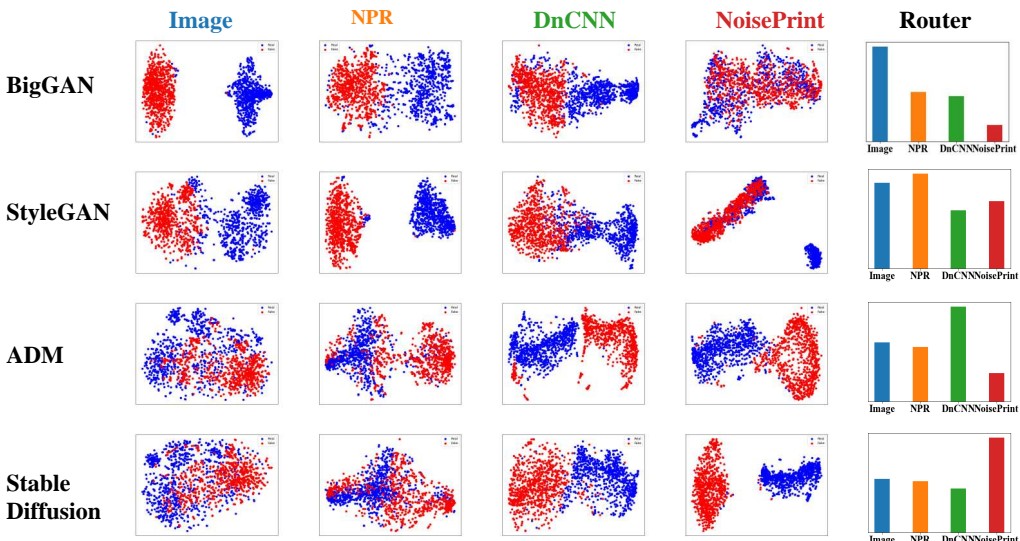

Figure 4: T-SNE visualization of features extracted by the classifier (Van Der Maaten, 2014). Blue and red represent the features of real images and fake images, respectively. The rightmost column shows the distribution bar chart of the selected different features when facing different forgery types.

StyleGAN, while DnCNN and NoisePrint delineate boundaries for ADM and Stable Diffusion. Our method adeptly selects the best features for classifying the current forgery type.

Table 4: Performance of different combinations of low-level information not used in the main text.

| Image | NPR | DnCNN | NoisePrint | Acc. | A.P. |
|-------|-----|-------|-----------|------|------|
| ✓ | | | | 85.3 | 91.8 |
| | ✓ | | | 84.6 | 91.4 |
| | | ✓ | | 83.9 | 89.6 |
| | | | ✓ | 85.1 | 90.1 |
| ✓ | ✓ | | | 89.1 | 93.2 |
| ✓ | ✓ | ✓ | | 91.3 | 95.1 |
| ✓ | ✓ | ✓ | ✓ | **93.3** | **98.4** |

Table 5: Performance of different combinations of model compoents.

| LE | LIIA | CLA | DFS | Acc. | A.P. |
|----|------|-----|-----|------|------|
| | | | | 80.8 | 87.6 |
| ✓ | | | | 89.0 | 93.7 |
| ✓ | ✓ | | | 91.7 | 96.0 |
| ✓ | | ✓ | | 90.6 | 95.3 |
| ✓ | ✓ | ✓ | | 92.8 | 97.8 |
| ✓ | ✓ | ✓ | ✓ | **93.3** | **98.4** |

**Core model components.** Tab. 5 presents the ablation study of our proposed model components: Lora Expert (LE), Cross-Low-level Attention (CLA), Low-level Information Interaction Adapter (LIIA), and Dynamic Feature Selection (DFS). Utilizing individual components and various combinations enhances the model's generalization performance on the test set. By employing all components, our method achieves improvements of 12.5% in Acc. and 10.8% in A.P. compared to using only low-level information and Image as input, followed by late fusion and fine-tuning the fully connected layer. To further illustrate the effectiveness of our fusion strategy, we visualize the Class Activation Map (CAM) for images with different forgery types and low-level information using the CAM method from (Zhou et al., 2016), shown in Fig. 5 in the Appendix. The results indicate that different low-level information highlights distinct regions for the same forgery type, and our fusion method effectively combines these focus regions to better identify hidden forgery clues in the images.

# 6 CONCLUSION

In this paper, we have discovered the advantage of various low-level features in enhancing the generalization capability of AI-generated image detection. We presents the Adaptive Low-level Experts Injection (ALEI) framework, which enhances AI-generated image detection through low-level features. By utilizing Lora Experts, our transformer-based approach learns from these features, merging them via a Cross-Low-level Attention layer. We introduce a Low-level Information Adapter to maintain the backbone's modeling ability and employ Dynamic Feature Selection to optimize feature selection for current images. Our method achieved state-of-the-art results on multiple datasets, demonstrating improved generalization in detecting AI-generated images.

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

# A APPENDIX

## A.1 MORE IMPLEMENTATION DETAILS

**Testing datasets.** In the main text, we used three datasets, **CNNDetectionBenchmark (Wang et al., 2020)**, **GANGenDetectionBenchmark (Tan et al., 2024)**, **UniversalFakeDetectBenchmark (Ojha et al., 2023)** and **AIGCDetectBenchmark (Zhong et al., 2023)**, to evaluate the generalization of our method across different types of forgeries. The following provides a more detailed description of these datasets:

- **CNNDetectionBenchmark (Wang et al., 2020):** This dataset includes fake images generated by various GAN methods such as ProGAN (Karras et al., 2018), StyleGAN (Karras et al., 2019), StyleGAN2 (Karras et al., 2020), BigGAN (Brock et al., 2018), CycleGAN (Zhu et al., 2017), StarGAN (Choi et al., 2018), GauGAN (Park et al., 2019), and DeepFake (Rossler et al., 2019). It also contains real images randomly selected from six datasets: LSUN (Yu et al., 2015), ImageNet (Deng et al., 2009), CelebA (Liu et al., 2015), CelebA-HQ (Karras et al., 2018), COCO (Lin et al., 2014), and FaceForensics++ (Rossler et al., 2019). This dataset is commonly used in early AIGC detection work.

- **GANGenDetectionBenchmark (Tan et al., 2024):** To better evaluate the generalization of our detection method on GAN-generated images, we follow (Tan et al., 2023a) and extend our evaluation with images generated by 9 additional GAN models. Each GAN model includes 4K test images, with an equal number of real and fake images.

- **UniversalFakeDetectBenchmark (Ojha et al., 2023):** This dataset includes test sets from diffusion methods such as ADM (Dhariwal and Nichol, 2021), DALL-E (Ramesh et al., 2021), LDM (Rombach et al., 2022), and Glide (Nichol et al., 2022). Variants of these methods are also considered for LDM and Glide. Real image datasets are drawn from LAION (Schuhmann et al., 2021) and ImageNet (Deng et al., 2009).

- **AIGCDetectBenchmark (Zhong et al., 2023):** Similar to cnndetection, this dataset collects fake images generated by seven GAN-based models and real images from the same sources. Additionally, it incorporates whichfaceisreal (WFIR) and GenImage (Zhu et al., 2023), collecting images from seven diffusion models.

**Implementation details.** For the LoRA expert module we use, we set $\alpha = 8$ and $r = 4$. As mentioned in the main text, these Lora experts are trained individually for each type of low-level information. The training steps are consistent with the implementation details in the main text. For the low-level encoder part, we also follow the same pre-training setup as in the main text, where the extracted features are trained using a classification head and cross-entropy loss to ensure that the features extracted from the low-level information are optimal for our classification task. We insert our Cross-Low-level attention layer and Low-level Information Adapter only at one-quarter, one-half, three-quarters, and the final layer of the pre-trained transformer backbone we use. We provide the code for reproducing our experiments in the supplementary materials, and more implementation details can be found in the code.

## A.2 MORE EXPERIMENTAL RESULTS

**Comparison on testing datasets.** The raw experimental data used to plot Fig. 1 and for the analysis in the methods section is presented in Tab. 6. Tab. 7 evaluate the Acc. and A.P. metrics on CNNDetection. Our method achieves excellent results compared to multiple baseline methods and yields comparable results with the current state-of-the-art methods NPR (Liu et al., 2023a) and FAFormer (Tan et al., 2023a). Specifically, our method improves Acc. by 3.4% and 0.1% compared to (Liu et al., 2023a) and (Tan et al., 2023a), respectively. For the StyleGAN, where (Liu et al., 2023a) performs poorly, and the BigGAN, where (Tan et al., 2023a) underperforms, our method improves the average accuracy by 10.7% and 7.0%, respectively. This demonstrates that our method, by incorporating multiple low-level information, uniformly enhances the detection performance across different GAN generation methods. Tab. 8, Tab. 9 and Tab. 10 are the complete versions of Tab. 1, Tab. 2 and Tab. 3 presented in the main text, respectively. They include more baseline method comparisons and additional test results on more datasets. Tab. 11 presents the results of some combinations of low-level information

Table 6: The detection accuracy comparison between different low-level information and fusion method. Among all detectors, the best result and the second-best result are denoted in boldface and underlined, respectively.

| Generator | Image | SRM | LNP | NPR | Bayar | DnCNN | Noiseprint | EarlyFusion | LateFusion | NPR(ResNet50) | Ours |
|---|---|---|---|---|---|---|---|---|---|---|---|
| ProGAN | 99.49 | 98.38 | 99.18 | **100.00** | 97.15 | 98.28 | 99.88 | 98.51 | 99.95 | 99.96 | **100.00** |
| StyleGAN | 89.45 | 79.00 | 69.22 | 96.59 | 77.85 | 83.68 | 82.69 | 83.99 | 99.12 | 97.28 | **98.35** |
| BigGAN | **96.95** | 82.23 | 88.33 | 86.13 | 70.28 | 81.40 | 72.53 | 75.88 | 87.78 | 85.88 | 94.51 |
| CycleGAN | **98.59** | 50.91 | 74.11 | 83.17 | 84.44 | 86.45 | 75.85 | 66.50 | 98.05 | 95.12 | 97.03 |
| StarGAN | 99.57 | 96.42 | 99.22 | 98.05 | 99.50 | 95.35 | **100.00** | 99.87 | 99.92 | 97.32 | **100.00** |
| GauGAN | 97.92 | 69.78 | 83.52 | 84.51 | 53.59 | 71.12 | 52.84 | 63.47 | 84.69 | **97.99** | 95.19 |
| StyleGAN2 | 91.71 | 77.52 | 73.38 | 96.53 | 81.15 | 79.75 | 87.18 | 78.79 | 96.61 | **99.56** | 98.88 |
| whichfaceisreal | 83.25 | 51.95 | 50.00 | 70.30 | 50.00 | 45.85 | **90.45** | 51.95 | 71.30 | 50.35 | 75.71 |
| ADM | 77.78 | 89.61 | 82.54 | 68.88 | 89.47 | **92.26** | 79.72 | 57.19 | 87.05 | 71.30 | 88.43 |
| Glide | 84.99 | 93.58 | 75.21 | 86.25 | 90.14 | **93.97** | 74.70 | 39.67 | 88.41 | 94.11 | 91.53 |
| Midjourney | 58.14 | 51.14 | 50.59 | 86.39 | 50.00 | 87.23 | **93.58** | 55.34 | 91.33 | 74.30 | 91.56 |
| SDv1.4 | 74.29 | 50.02 | 50.20 | 86.12 | 50.00 | 81.24 | 91.18 | 56.70 | 89.83 | 69.43 | **93.28** |
| SDv1.5 | 74.40 | 49.96 | 49.96 | 85.88 | 50.00 | 81.29 | 91.14 | 56.52 | 89.96 | 69.51 | **93.38** |
| VQDM | 85.43 | 77.27 | 68.51 | 69.94 | 87.79 | **93.07** | 87.43 | 57.25 | 91.23 | 80.80 | 90.94 |
| wukong | 77.29 | 50.04 | 50.14 | 78.88 | 50.00 | 76.67 | **89.52** | 56.87 | 83.45 | 61.97 | 89.46 |
| DALLE2 | 75.90 | 92.30 | 83.05 | 76.00 | 94.55 | **95.00** | 93.40 | 33.45 | 92.40 | 93.25 | 93.32 |
| **Average** | 85.32 | 72.51 | 71.70 | 84.60 | 73.49 | 83.91 | 85.13 | 64.50 | 90.69 | 83.63 | **93.29** |

not utilized in the main text, demonstrating that our framework can effectively integrate other low-level information that may possess generalization capabilities. Tab. 12 displays the parameters related to the model's performance and inference efficiency, indicating that our method achieves a balance between performance and efficiency.

Table 7: Cross-GAN-Sources Evaluation on the test set of CNNDetection (Wang et al., 2020). Partial results from (Liu et al., 2023a; Tan et al., 2023a).

| Method | ProGAN | | StyleGAN | | StyleGAN2 | | BigGAN | | CycleGAN | | StarGAN | | GauGAN | | Deepfake | | Mean | |
|---|---|---|---|---|---|---|---|---|---|---|---|---|---|---|---|---|---|---|
| | Acc. | A.P. | Acc. | A.P. | Acc. | A.P. | Acc. | A.P. | Acc. | A.P. | Acc. | A.P. | Acc. | A.P. | Acc. | A.P. | Acc. | A.P. |
| CNNDetection | 91.4 | 99.4 | 63.8 | 91.4 | 76.4 | 97.5 | 52.9 | 73.3 | 72.7 | 88.6 | 63.8 | 90.8 | 63.9 | 92.2 | 51.7 | 62.3 | 67.1 | 86.9 |
| Frank | 90.3 | 85.2 | 74.5 | 72.0 | 73.1 | 71.4 | 88.7 | 86.0 | 75.5 | 71.2 | 99.5 | 99.5 | 69.2 | 77.4 | 60.7 | 49.1 | 78.9 | 76.5 |
| Durall | 81.1 | 74.4 | 54.4 | 52.6 | 66.8 | 62.0 | 60.1 | 56.3 | 69.0 | 64.0 | 98.1 | 98.1 | 61.9 | 57.4 | 50.2 | 50.0 | 67.7 | 64.4 |
| Patchfor | 97.8 | 100.0 | 82.6 | 93.1 | 83.6 | 98.5 | 64.7 | 69.5 | 74.5 | 87.2 | 100.0 | 100.0 | 57.2 | 55.4 | 85.0 | 93.2 | 80.7 | 87.1 |
| F3Net | 99.4 | 100.0 | 92.6 | 99.7 | 88.0 | 99.8 | 65.3 | 69.9 | 76.4 | 84.3 | 100.0 | 100.0 | 58.1 | 56.7 | 63.5 | 78.8 | 80.4 | 86.2 |
| SelfBlend | 58.8 | 65.2 | 50.1 | 47.7 | 48.6 | 47.4 | 51.1 | 51.9 | 59.2 | 65.3 | 74.5 | 89.2 | 59.2 | 65.5 | **93.8** | **99.3** | 61.9 | 66.4 |
| GANDetection | 82.7 | 95.1 | 74.4 | 92.9 | 69.9 | 87.9 | 76.3 | 89.9 | 85.2 | 95.5 | 68.8 | 99.7 | 61.4 | 75.8 | 60.0 | 83.9 | 72.3 | 90.1 |
| BiHPF | 90.7 | 86.2 | 76.9 | 75.1 | 76.2 | 74.7 | 84.9 | 81.7 | 81.9 | 78.9 | 94.4 | 94.4 | 69.5 | 78.1 | 54.4 | 54.6 | 78.6 | 77.9 |
| FrePGAN | 99.0 | 99.9 | 80.7 | 89.6 | 84.1 | 98.6 | 69.2 | 71.1 | 71.1 | 74.4 | 99.9 | 100.0 | 60.3 | 71.7 | 70.9 | 91.9 | 79.4 | 87.2 |
| LGrad | 99.9 | 100.0 | 94.8 | 99.9 | 96.0 | 99.9 | 82.9 | 90.7 | 85.3 | 94.0 | 99.6 | 100.0 | 72.4 | 79.3 | 58.0 | 67.9 | 86.1 | 91.5 |
| UnivFD | 99.7 | 100.0 | 89.0 | 98.7 | 83.9 | 98.4 | 90.5 | 99.1 | 87.9 | 99.8 | 91.4 | 100.0 | 89.9 | 100.0 | 80.2 | 90.2 | 89.1 | 98.3 |
| NPR | 99.8 | 100.0 | 96.3 | 99.8 | 97.3 | 100.0 | 87.5 | 94.5 | 95.0 | 99.5 | 99.7 | 100.0 | 86.6 | 88.8 | 77.4 | 86.2 | 92.5 | 96.1 |
| FAFormer | 99.8 | 100.0 | 87.7 | 97.4 | 91.1 | 99.3 | **98.9** | **99.9** | **99.9** | **100.0** | 100.0 | 100.0 | **99.9** | **100.0** | 89.4 | 97.3 | 95.8 | **99.2** |
| Ours | **100.0** | 100.0 | **98.4** | 100.0 | **98.9** | 100.0 | 94.5 | 98.8 | 97.0 | 99.9 | 100.0 | 100.0 | 95.2 | 98.9 | 83.4 | 88.2 | **95.9** | 98.2 |

Table 8: The detection accuracy comparison between our approach and baselines. Among all detectors, the best result and the second-best result are denoted in boldface and underlined, respectively.

| Generator | CNNDet | FreDect | Fusing | GramNet | LNP | LGrad | DIRE-G | DIRE-D | UnivFD | PatchCraft | Ours |
|---|---|---|---|---|---|---|---|---|---|---|---|
| ProGAN | **100.00** | 99.36 | **100.00** | 99.99 | 99.95 | 99.83 | 95.19 | 52.75 | 99.81 | **100.00** | **100.00** |
| StyleGAN | 90.17 | 78.02 | 85.20 | 87.05 | 92.64 | 91.08 | 83.03 | 51.31 | 84.93 | 92.77 | **98.35** |
| BigGAN | 71.17 | 81.97 | 77.40 | 67.33 | 88.43 | 85.62 | 70.12 | 49.70 | 95.08 | **95.80** | 94.51 |
| CycleGAN | 87.62 | 78.77 | 87.00 | 86.07 | 79.07 | 86.94 | 74.19 | 49.58 | **98.33** | 70.17 | 97.03 |
| StarGAN | 94.60 | 94.62 | 97.00 | 95.05 | **100.00** | 99.27 | 95.47 | 46.72 | 95.75 | 99.97 | **100.00** |
| GauGAN | 81.42 | 80.57 | 77.00 | 69.35 | 79.17 | 78.46 | 67.79 | 51.23 | **99.47** | 71.58 | 95.19 |
| StyleGAN2 | 86.91 | 66.19 | 83.30 | 87.28 | 93.82 | 85.32 | 75.31 | 51.72 | 74.96 | 89.55 | **98.88** |
| whichfaceisreal | **91.65** | 50.75 | 66.80 | 86.80 | 50.00 | 55.70 | 58.05 | 53.30 | 86.90 | 85.80 | 75.71 |
| ADM | 60.39 | 63.42 | 49.00 | 58.61 | 83.91 | 67.15 | 75.78 | **98.25** | 66.87 | 82.17 | 88.43 |
| Glide | 58.07 | 54.13 | 57.20 | 54.50 | 83.50 | 66.11 | 71.75 | **92.42** | 62.46 | 83.79 | 91.53 |
| Midjourney | 51.39 | 45.87 | 52.20 | 50.02 | 69.55 | 65.35 | 58.01 | 89.45 | 56.13 | 90.12 | **91.56** |
| SDv1.4 | 50.57 | 38.79 | 51.00 | 51.70 | 89.33 | 63.02 | 49.74 | 91.24 | 63.66 | **95.38** | 93.28 |
| SDv1.5 | 50.53 | 39.21 | 51.40 | 52.16 | 88.81 | 63.67 | 49.83 | 91.63 | 63.49 | **95.30** | 93.38 |
| VQDM | 56.46 | 77.80 | 55.10 | 52.86 | 85.03 | 72.99 | 53.68 | **91.90** | 85.31 | 88.91 | 90.94 |
| wukong | 51.03 | 40.30 | 51.70 | 50.76 | 86.39 | 59.55 | 54.46 | 90.90 | 70.93 | **91.07** | 89.46 |
| DALLE2 | 50.45 | 34.70 | 52.80 | 49.25 | 92.45 | 65.45 | 66.48 | 92.45 | 50.75 | **96.60** | 93.32 |
| Average | 69.73 | 63.28 | 67.63 | 68.43 | 85.28 | 75.11 | 67.90 | 72.70 | 76.80 | 89.85 | **93.29** |

**Robustness Tests.** In real-world applications, images spread on public platforms may undergo various common image processing techniques like JPEG compression. Therefore, it is important to evaluate the performance of the detector when handling distorted images. We adopt three common image distortions, including JPEG compression (quality factor QF=95), Gaussian blur ($\sigma = 1$), and image downsampling, where the image size is reduced to a quarter of its original size ($r = 0.5$). Consistent with previous methods (Wang et al., 2020) and (Zhong et al., 2023), we augment the training set using the aforementioned image distortion methods and test on the AIGCDetectBenchmark test set processed with these distortion methods. The results are presented in Tab. 13. The results show that

Table 9: Cross-GAN-Sources Evaluation on the GANGenDetection (Tan et al., 2024). Partial results from (Tan et al., 2023a)

| Method | AttGAN | | BEGAN | | CramerGAN | | InfoMaxGAN | | MMDGAN | | RelGAN | | S3GAN | | SNGAN | | STGAN | | Mean | |
|---|---|---|---|---|---|---|---|---|---|---|---|---|---|---|---|---|---|---|---|---|
| | Acc. | A.P. | Acc. | A.P. | Acc. | A.P. | Acc. | A.P. | Acc. | A.P. | Acc. | A.P. | Acc. | A.P. | Acc. | A.P. | Acc. | A.P. | Acc. | A.P. |
| CNNDet | 51.1 | 83.7 | 50.2 | 44.9 | 81.5 | 97.5 | 71.1 | 94.7 | 72.9 | 94.4 | 53.3 | 82.1 | 55.2 | 66.1 | 62.7 | 90.4 | 63.0 | 92.7 | 62.3 | 82.9 |
| Frank | 65.0 | 74.4 | 39.4 | 39.9 | 31.0 | 36.0 | 41.1 | 41.0 | 38.4 | 40.5 | 69.2 | 96.2 | 69.7 | 81.9 | 48.4 | 47.9 | 25.4 | 34.0 | 47.5 | 54.7 |
| Durall | 39.9 | 38.2 | 48.2 | 30.9 | 60.9 | 67.2 | 50.1 | 51.7 | 59.5 | 65.5 | 80.0 | 88.2 | 87.3 | 97.0 | 54.8 | 58.9 | 62.1 | 72.5 | 60.3 | 63.3 |
| Patchfor | 68.0 | 92.9 | 97.1 | 100.0 | 97.8 | 99.9 | 93.6 | 98.2 | 97.9 | 100.0 | 99.6 | 100.0 | 66.8 | 68.1 | 97.6 | 99.8 | 92.7 | 99.8 | 90.1 | 95.4 |
| F3Net | 85.2 | 94.8 | 87.1 | 97.5 | 89.5 | 99.8 | 67.1 | 83.1 | 73.7 | 99.6 | 98.8 | 100.0 | 65.4 | 70.0 | 51.6 | 93.6 | 60.3 | 99.9 | 75.4 | 93.1 |
| SelfBlend | 63.1 | 66.1 | 56.4 | 59.0 | 75.1 | 82.4 | 79.0 | 82.5 | 68.6 | 74.0 | 73.6 | 77.8 | 53.2 | 53.9 | 61.6 | 65.0 | 61.2 | 66.7 | 65.8 | 69.7 |
| GANDet | 57.4 | 75.1 | 67.9 | 100.0 | 67.8 | 99.7 | 67.6 | 92.4 | 67.7 | 99.3 | 60.9 | 86.2 | 69.6 | 83.5 | 66.7 | 90.6 | 69.6 | 97.2 | 66.1 | 91.6 |
| LGrad | 68.6 | 93.8 | 69.9 | 89.2 | 50.3 | 54.0 | 71.1 | 82.0 | 57.5 | 67.3 | 89.1 | 99.1 | 78.5 | 86.0 | 78.0 | 87.4 | 54.8 | 68.0 | 68.6 | 80.8 |
| UnivFD | 78.5 | 98.3 | 72.0 | 98.9 | 77.6 | 99.8 | 77.6 | 98.9 | 77.6 | 99.7 | 78.2 | 98.7 | 85.2 | 98.1 | 77.6 | 98.7 | 74.2 | 97.8 | 77.6 | 98.8 |
| NPR | 83.0 | 96.2 | 99.0 | 99.8 | 98.7 | 99.0 | 94.5 | 98.3 | 98.6 | 99.0 | 99.6 | 100.0 | 79.0 | 80.0 | 88.8 | 97.4 | 98.0 | 100.0 | 93.2 | 96.6 |
| Ours | **86.2** | **97.8** | **100.0** | **100.0** | **100.0** | **100.0** | **98.6** | **99.9** | **99.3** | **99.8** | **100.0** | **100.0** | 83.0 | 87.0 | 90.4 | 98.7 | **100.0** | **100.0** | **95.3** | **98.1** |

Table 10: Cross-Diffusion-Sources Evaluation on the diffusion test set of UniversalFakeDetect (Ojha et al., 2023). Partial results from (Liu et al., 2023a; Tan et al., 2023a).

| Method | DALLE | | Glide_100_10 | | Glide_100_27 | | Glide_50_27 | | ADM | | LDM_100 | | LDM_200 | | LDM_200_cfg | | Mean | |
|---|---|---|---|---|---|---|---|---|---|---|---|---|---|---|---|---|---|---|
| | Acc. | A.P. | Acc. | A.P. | Acc. | A.P. | Acc. | A.P. | Acc. | A.P. | Acc. | A.P. | Acc. | A.P. | Acc. | A.P. | Acc. | A.P. |
| CNNDet | 51.8 | 61.3 | 53.3 | 72.9 | 53.0 | 71.3 | 54.2 | 76.0 | 54.9 | 66.6 | 51.9 | 63.7 | 52.0 | 64.5 | 51.6 | 63.1 | 52.8 | 67.4 |
| Frank | 57.0 | 62.5 | 53.6 | 44.3 | 50.4 | 40.8 | 52.0 | 42.3 | 53.4 | 52.5 | 56.6 | 51.3 | 56.4 | 50.9 | 56.5 | 52.1 | 54.5 | 49.6 |
| Durall | 55.9 | 58.0 | 54.9 | 52.3 | 48.9 | 46.9 | 51.7 | 49.9 | 40.6 | 42.3 | 62.0 | 62.6 | 61.7 | 61.7 | 58.4 | 58.5 | 54.3 | 54.0 |
| Patchfor | 79.8 | 99.1 | 87.3 | 99.7 | 82.8 | 99.1 | 84.9 | 98.8 | 74.2 | 81.4 | 95.8 | 99.8 | 95.6 | 99.9 | 94.0 | 99.8 | 86.8 | 97.2 |
| F3Net | 71.6 | 79.9 | 88.3 | 95.4 | 87.0 | 94.5 | 88.5 | 95.4 | 69.2 | 70.8 | 74.1 | 84.0 | 73.4 | 83.3 | 80.7 | 89.1 | 79.1 | 86.5 |
| SelfBlend | 52.4 | 51.6 | 58.8 | 63.2 | 59.4 | 64.1 | 64.2 | 68.3 | 58.3 | 63.4 | 53.0 | 54.0 | 52.6 | 51.9 | 51.9 | 52.6 | 56.3 | 58.7 |
| GANDet | 67.2 | 83.0 | 51.2 | 52.6 | 51.1 | 51.9 | 51.7 | 53.5 | 49.6 | 49.0 | 54.7 | 65.8 | 54.9 | 65.9 | 53.8 | 58.9 | 54.3 | 60.1 |
| LGrad | 88.5 | 97.3 | 89.4 | 94.9 | 87.4 | 93.2 | 90.7 | 95.1 | 86.6 | 100.0 | 94.8 | 99.2 | 94.2 | 99.1 | 95.9 | 99.2 | 90.9 | 97.2 |
| UnivFD | 89.5 | 96.8 | 90.1 | 97.0 | 90.7 | 97.2 | 91.1 | 97.4 | 75.7 | 85.1 | 90.5 | 97.0 | 90.2 | 97.1 | 77.3 | 88.6 | 86.9 | 94.5 |
| NPR | 94.5 | 99.5 | 98.2 | 99.8 | **97.8** | **99.7** | 98.2 | 99.8 | 75.8 | 81.0 | 99.3 | 99.9 | **99.1** | **99.9** | **99.0** | **99.8** | 95.2 | 97.4 |
| FAFormer | **98.8** | **99.8** | 94.2 | 99.2 | 94.4 | 99.1 | 94.7 | 99.4 | 76.1 | 92.0 | 98.7 | 99.9 | 98.6 | 99.8 | 94.9 | 99.1 | 93.8 | 95.5 |
| Ours | 97.7 | 99.7 | 97.9 | 99.2 | 97.3 | 99.1 | **98.6** | **99.9** | 90.1 | 96.4 | 99.5 | 99.9 | 98.9 | 99.3 | 98.5 | 99.5 | **97.3** | **99.1** |

compared to previous methods, our method achieves better robustness, outperforming the current best methods by 8.04%, 6.13%, and 7.27% in robustness tests for JPEG compression, Gaussian blur, and image downsampling, respectively. Fig. 6 visualizes the low-level information we use, including high-level images, before and after these operations. For low-level information, these operations partially affect it. However, due to our robust training and the introduction of high-level images along with multiple low-level features, our method's robustness is effectively enhanced.

**Transfer to other pretraining methods.** To further demonstrate the generality of our proposed method, we analyze its performance when combined with different architectures and pretraining strategies. Tab. 14 shows the Acc. and A.P. metrics for different pretrained models and various backbones. By comparing the performance with and without our method, we verify the effectiveness of incorporating low-level information and using our fusion architecture under different pretraining frameworks. This significantly improves the generalization of these methods for detecting synthetic images.

Table 11: Robustness performance(Acc.) on different baselines and our method. the best result and the second-best result are denoted in boldface and underlined, respectively

| Image | SRM | LNP | Bayar | Acc. | A.P. |
|---|---|---|---|---|---|
| ✓ | | | | 85.3 | 91.8 |
| | ✓ | | | 72.5 | 84.4 |
| | | ✓ | | 71.7 | 83.2 |
| | | | ✓ | 73.5 | 87.9 |
| ✓ | ✓ | | | 87.8 | 92.4 |
| ✓ | ✓ | ✓ | | 89.3 | 93.1 |
| ✓ | | ✓ | ✓ | 88.1 | 92.7 |
| ✓ | ✓ | | ✓ | 90.4 | 93.0 |
| ✓ | ✓ | ✓ | ✓ | **90.7** | **95.6** |

Table 12: Analysis of the model's parameters, performance, and inference efficiency.

| Model | Total Params (M) | Inference Time (s) | Mean Acc. |
|---|---|---|---|
| CNNDet(ResNet50) | **23.51** | **0.004** | 69.7 |
| Lgrad(ResNet50) | 46.56 | 0.012 | 75.1 |
| UniFD(CLIP-L/14) | 332.32 | 0.327 | 76.8 |
| NPR(CLIP-L/14) | 332.32 | 0.329 | 84.6 |
| Ours(ResNet50) | 94.04 | 0.014 | 91.7 |
| Ours(CLIP-L/14) | 366.19 | 0.391 | **93.3** |

Table 13: Robustness performance(Acc.) on different baselines and our method. the best result and the second-best result are denoted in bold-face and underlined, respectively

| Detector | JPEG | Downsampling | Blur |
|---|---|---|---|
| CNNDetction | 64.03 | 58.85 | 68.39 |
| FreDect | 66.95 | 35.84 | 65.75 |
| Fusing | 62.43 | 50.00 | 68.09 |
| GramNet | 65.47 | 60.30 | 68.63 |
| LNP | 53.56 | 63.28 | 65.88 |
| LGrad | 51.55 | 60.86 | 71.73 |
| DIRE-G | 66.49 | 56.09 | 64.00 |
| DIRE-D | 70.27 | 62.26 | 70.46 |
| UnivFD | 74.10 | 70.87 | 70.31 |
| Patchcraft | 72.48 | 78.36 | 75.99 |
| Ours | **80.52** | **84.49** | **83.26** |

Table 14: Analysis of different architectures and pretraining strategies.

| Arch | Pretrain | w/Ours | Acc. | A.P. |
|---|---|---|---|---|
| ViT-B | ImageNet (Deng et al., 2009) | × | 71.7 | 88.5 |
| | | ✓ | **85.4** | **93.6** |
| ViT-L | ImageNet (Deng et al., 2009) | × | 76.2 | 89.0 |
| | | ✓ | **89.7** | **94.2** |
| ViT-B | SAM (Kirillov et al., 2023) | × | 63.3 | 81.2 |
| | | ✓ | **80.1** | **89.9** |
| ViT-L | SAM (Kirillov et al., 2023) | × | 66.6 | 82.4 |
| | | ✓ | **81.1** | **86.8** |
| ViT-B | CLIP (Radford et al., 2021) | × | 72.5 | 85.1 |
| | | ✓ | **86.8** | **93.6** |
| ViT-L | CLIP (Radford et al., 2021) | × | 76.8 | 90.2 |
| | | ✓ | **93.3** | **98.4** |

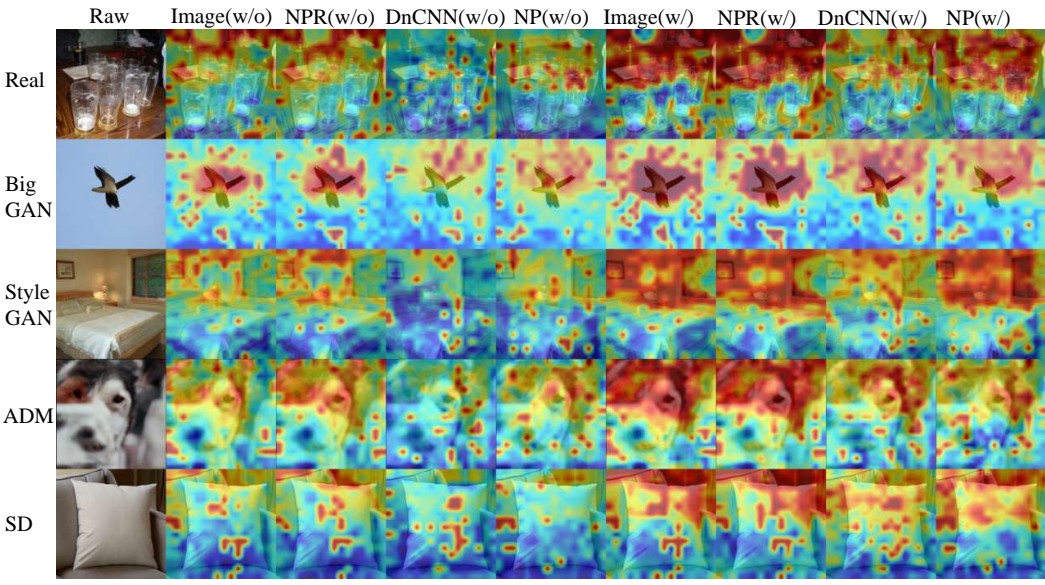

Figure 5: Visualization of the Class Activation Map (CAM) corresponding to different forgery types and different low-level information (Zhou et al., 2016). Warmer colors indicate higher probabilities.

### A.3 BROADER IMPACTS AND LIMITATION

As AI-generated image detection methods continue to evolve, they aim to combat the growing influx of fake information and the constantly updating AIGC technologies. However, these methods may have unintended consequences in the realm of content moderation. Legitimate human-created content that resembles forgeries may be incorrectly identified as AI-generated images, while some highly realistic AI-generated images might be recognized by algorithms as genuine. This could impact the sharing of normal information based on image morphology. Further research and consideration are needed when applying this work to practical applications in content moderation.

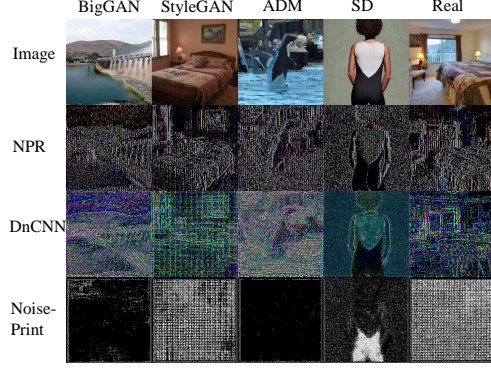

(a) The original image and low-level information.

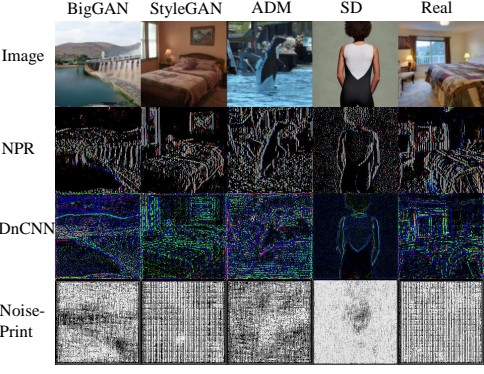

(b) The blurred image and low-level information.

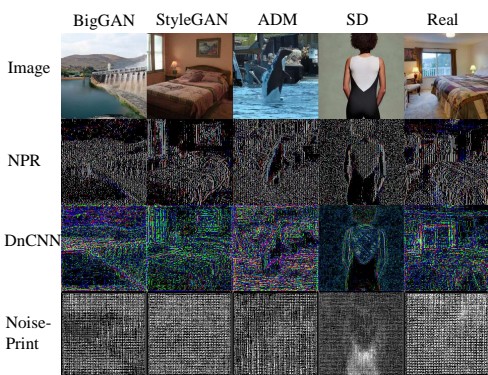

(c) The JPEG compressed image and low-level information.

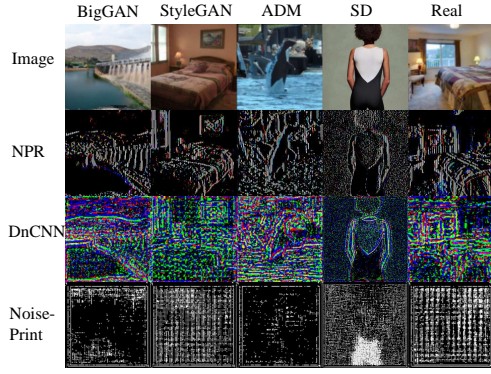

(d) The downsampled image and low-level information.

Figure 6: The visualization results of the image and low-level information.

