# OpenReview forum: "Exploring the Collaborative Advantage of Low-level Information on Generalizable AI-generateted Image Detection"
_ICLR.cc/2025/Conference — ICLR 2025 Conference Withdrawn Submission_

### Official Review · Reviewer_YJQF · 2024-11-03

**Soundness:** 3
**Presentation:** 3
**Contribution:** 2
**Rating:** 5
**Confidence:** 4

**Summary:**

In the current image detection task, the existing detection methods are based on various high-level information or low-level information to achieve. This paper proposes the Adaptive Low-level Experts Injection framework, which uses multiple low-level information pairs to detect the generated images. The framework is based on CLIP, uses LORA to extend the attention module, and uses the cross-attention mechanism to integrate multiple information. Training on four categories of ProGAN datasets, the method achieves SOTA effect on data including GAN and Diffusion models.

**Strengths:**

1. The network structure of this paper is novel. In this paper, CLIP network is used as the backbone network, and a Low-level Information Adapter module is designed to input low-level information into the CLIP network, which mainly extracts semantic information, so as to enhance the embedding of original information.
2. In this paper, a large number of experimental accuracy results on the generated image dataset are presented, including the mainstream generation methods based on GAN and stable diffusion. The data are detailed and the method is relatively solid.
3. In general, similar to the idea of using multi-expert models to detect fake images, it is relatively valuable to study.

**Weaknesses:**

1. Lack of necessary explanation: In this article, the judgment method is combined with various underlying information. The existing single underlying information method in the article is explained. However, the lack of detailed comparison and explanation with the method that also uses multiple underlying information points out the shortcomings of the existing multi-information method to prove the effectiveness of the method. Also specify if this is the first method of this type.
2. This paper introduces a variety of low-level information, such as SRM filters, DIRE, etc. However, NPR, DnCNN, and NoisePrint are selected in the model. According to the description in the paper, the model's generalization performance is enhanced with more information, and data is needed to explain the reasons for selecting these three types of information.
3. In the detection task of diffusion model, the effect is inferior to PatchCraft, and the possible reasons need to be explained.

**Questions:**

1. Lack of necessary explanation: In this article, the judgment method is combined with various underlying information. The existing single underlying information method in the article is explained. However, the lack of detailed comparison and explanation with the method that also uses multiple underlying information points out the shortcomings of the existing multi-information method to prove the effectiveness of the method. Also specify if this is the first method of this type.
2. This paper introduces a variety of low-level information, such as SRM filters, DIRE, etc. However, NPR, DnCNN, and NoisePrint are selected in the model. According to the description in the paper, the model's generalization performance is enhanced with more information, and data is needed to explain the reasons for selecting these three types of information.
3. Your model is not as effective as PatchCraft in detecting diffusion models. Possible causes need to be explained.
4. The resolution used in training is 224*224, is it the same resolution when reasoning? As for the diffusion model, this model is better generated at large resolution. At what resolution does your reasoning work?

---

### Official Review · Reviewer_7NWd · 2024-11-03

**Soundness:** 2
**Presentation:** 2
**Contribution:** 2
**Rating:** 3
**Confidence:** 5

**Summary:**

This paper proposed the Adaptive Low-level Experts Injection framework and developed a Low-level Information Adapter that interacts with the features extracted by the backbone. Experiments on several datasets show that the proposed model achieves state-of-the-art performance.

**Strengths:**

- The motivation of this paper is clear and the authors chose a straightforward but effective method to achieve the goal.
- The combination of low-level features, Lora Experts, and feature selection for AI-generateted Image Detection is interesting.

**Weaknesses:**

- What concerns me the most is whether the Lora Experts is truly effective and whether the experiments conducted by the authors are solid enough. The core issue of the AI-generated Image Detection task lies in the generalization of the model. However, from the experimental results in the tables, it can be observed that the algorithm does not perform well enough when facing cross-models, especially diffusion models.
- Moreover, it is better if the paper provides some failure cases in the visualization figure and explains the reason why these cases happen.
- Some figures in the paper are not clear enough, it is recommended to utilize figures in PDF or eps format. Moreover, it is better to unify fonts in all figures.
- Please unify the format of references. At least ensure that the citation formats of conferences and journals are consistent.
- The sources of citations in this paper should be corrected. For example, “Dire for diffusion-generated image detection” is from CVPR2023 rather than arXiv.
- There are some typos in this paper, such as “Many work (works) (Zhao et al., 2023; Peng et al., 2021; Yuan et al., 2021) suggests …… (4.3 LOW-LEVEL INFORMATION INTERACTION ADAPTER)”.

**Questions:**

- Please standardize the capitalization of English letters in the references. Many abbreviations of proper nouns are incorrect, such as Stargan (StarGAN).
- We usually use "generated" rather than "generated".

---

### Official Review · Reviewer_Pf39 · 2024-11-04

**Soundness:** 2
**Presentation:** 2
**Contribution:** 2
**Rating:** 5
**Confidence:** 3

**Summary:**

1.	This paper proposes the Adaptive Low-level Experts Injection (ALEI) framework for AI-Generated image detection.
2.	Extensive experiments demonstrate that ALEI achieves state-of-the-art results on multiple benchmarks.

**Strengths:**

The method is effective in many benchmarks.

**Weaknesses:**

1.	Limited novelty: This method is mainly a simple fusion of high-level information and low-level information, which has been introduced into AI-Generated image detection in [1]. Moreover, high-level information has been introduced in [2] and low-level information has been introduced in [3].
2.	The concept of utilizing multiple LoRAs is quite prevalent, as highlighted in [4], where their application in AI-generated image detection is discussed.
3.	ALEI is much bigger than existing work such as PatchCraft that uses just few convolution layers as the classifier. This makes one wonder whether the performance increase claimed in the article is due to the increased FLOPS/PARAMS or the ALEI itself.
4.	Lack of a lot of experiments. Lack of experimental results for benchmarks such as GenImage, DiffusionForensis, etc.
5.	Performance evaluation. Table 1 indicates that PatchCraft [69] outperforms the proposed method across several generators，especially in diffusion-based generators.
[1]. A Sanity Check for AI-generated Image Detection
[2]. Towards Universal Fake Image Detectors that Generalize Across Generative Models
[3]. PatchCraft: Exploring Texture Patch for Efficient AI-generated Image Detection
[4]. MoE-FFD: Mixture of Experts for Generalized and Parameter-Efficient Face Forgery Detection

**Questions:**

See weakness

---

### Official Review · Reviewer_1NAQ · 2024-11-04

**Soundness:** 3
**Presentation:** 4
**Contribution:** 2
**Rating:** 5
**Confidence:** 4

**Summary:**

This paper instroduces Adaptive Low-level Experts Injection (ALEI) framework to investigate the generalization issue in AI-Generated image detection. The authors claim that that integrating diverse low-level information helps overcome the limitations of generalizing to unseen generative models. Besides, Cross-Low-level Attention and Dynamic Feature Selection are used to fuse low level features and to select suitable features dynamically.

**Strengths:**

1. This paper uses low-level features to address the issue of limited generation in AI-Generated image detection and gives some insights into it. Also, the experiments and analysis of different types of low-level information shows that each type has distinct contribution of detection.
2. The cross-low-level attention layer integrates different low-level features without losing their unique contributions, helping avoid the pitfalls of simple feature fusion.

**Weaknesses:**

1. The ALEI uses LoRA experts to integrate low-level features, I wonder what are the criteria used to determine the number of experts? Is this method sensitive to the number of experts?
2. In dynamic feature selection, how does the model ensure the reliability of this selection, especially for forgeries that may lie in the overlapping space of multiple generative models? Such as the potential risk of feature redundancy or competition between modalities during selection.
3. The model is trained on ProGAN firstly, and uses further training with the fusion module. How does the two-stage training process impact the final generalization ability? Would joint training perform better? Or is there a risk of overfitting due to low-level and high-level features competition?

**Questions:**

See weaknesses.

---

### Note · Authors · 2024-11-13

I have read and agree with the venue's withdrawal policy on behalf of myself and my co-authors.